# Methodology of Purification of Inactivated Cell-Culture-Grown SARS-CoV-2 Using Size-Exclusion Chromatography

**DOI:** 10.3390/vaccines10060949

**Published:** 2022-06-15

**Authors:** Anastasia A. Kovpak, Anastasia N. Piniaeva, Oleg A. Gerasimov, Irina O. Tcelykh, Mayya Y. Ermakova, Anna N. Zyrina, Dmitry V. Danilov, Yury Y. Ivin, Liubov I. Kozlovskaya, Aydar A. Ishmukhametov

**Affiliations:** 1Chumakov Federal Scientific Center for Research and Development of Immune-and-Biological Products of Russian Academy of Sciences, 108819 Moscow, Russia; pinyaeva_an@chumakovs.su (A.N.P.); gerasimov_oa@chumakovs.su (O.A.G.); tselih_io@chumakovs.su (I.O.T.); ermakova_mj@chumakovs.su (M.Y.E.); zyrina_an@chumakovs.su (A.N.Z.); danilov_dv@chumakovs.su (D.V.D.); ivin_uu@chumakovs.su (Y.Y.I.); kozlovskaya_li@chumakovs.su (L.I.K.); ishmukhametov@chumakovs.su (A.A.I.); 2Institute of Translational Medicine and Biotechnology, Sechenov Moscow State Medical University, 119991 Moscow, Russia

**Keywords:** vaccine, Size-Exclusion Chromatography, COVID-19, chromatography, SARS-CoV-2, resin

## Abstract

Various types of COVID-19 vaccines, including adenovirus, mRNA, and inactivated ones, have been developed and approved for clinical use worldwide. Inactivated vaccines are produced using a proven technology that is widely used for the production of vaccines for the prevention and control of infectious diseases, including influenza and poliomyelitis. The development of inactivated whole-virion vaccines commonly includes several stages: the production of cellular and viral biomass in cell culture; inactivation of the virus; filtration and ultrafiltration; chromatographic purification of the viral antigen; and formulation with stabilizers and adjuvants. In this study, the suitability of four resins for Size-Exclusion Chromatography was investigated for the purification of a viral antigen for the human COVID-19 vaccine.

## 1. Introduction

Severe acute respiratory syndrome coronavirus 2 (SARS-CoV-2) is an etiological agent of COVID-19 [1], classified as a pandemic by the WHO [2]. Rapid identification of the etiological agent, information on the genetic sequence of the virus, and free access to it led to the rapid availability of diagnostic tools and the development of prophylactic vaccines.

SARS-CoV-2 is an enveloped, single-stranded RNA virus belonging to the *Betacoronavirus* genus of the *Coronaviridae* family [3]. A spherical enveloped virion incorporates genomic RNA covered with nucleocapsid (N) proteins, while the main glycoprotein (S) forms trimeric spikes on the virion surface and two minor proteins the membrane (M), and the envelope (E) do not protrude from the membrane. The S protein consists of two subunits: S1 and S2. The receptor-binding domain (RBD) is located in the S1 subunit and determines the attachment of the virus to its receptor—angiotensin-converting enzyme 2 (hACE2) [4]. After attachment, the S protein is cleaved by host-cell proteases such as transmembrane serine protease 2 (TMPRSS2) and cathepsin, thereby inducing the fusion of viral and cellular membranes [5,6]. Moreover, RBD is the main target for neutralizing the antibodies of the host immune system. Therefore, it is considered an ideal target for vaccines that trigger the production of virus-neutralizing antibodies [5].

Various types of COVID-19 vaccines have already been developed and approved for clinical use worldwide. COVID-19 vaccines can be divided into four categories based on the different platforms for their production: (1) whole-virion viral vaccines, (2) protein-based vaccines, (3) viral vector vaccines, and (4) nucleic-acid-based vaccines. A weakened (attenuated) or inactivated coronavirus can be used in whole-virion viral vaccines [7]. Inactivated vaccines are produced using a proven technology that is widely used for the production of vaccines for the prevention and control of infectious diseases, including influenza and poliomyelitis [8,9,10,11].

The development of inactivated whole-virion viral vaccines includes several common stages: the production of cellular and viral biomass from virus-susceptible cells (for coronaviruses, it is the Vero cell line); inactivation of the virus by chemical compounds (commonly used formaldehyde or β-propiolactone) or physical factors (heat or UV radiation); filtration and ultrafiltration; chromatographic purification of the virus; and formulation of the vaccine with adjuvants and other compounds, if needed. The knowledge of purifications methods of the whole-virion coronavirus antigen is rather scarce. Most publications describe chromatographic methods for the purification of coronavirus proteins (nucleocapsid or spike) obtained in various expression systems. Often, these technologies involve affinity chromatography, which may not be applicable for purification of the whole-virion particles [12,13]. On the other hand, publications describing chromatographic methods for the purification of inactivated whole-virion antigens do not present the data in detail, or present them without comparison [14,15].

We developed a β-propiolactone-inactivated whole-virion vaccine CoviVac based on the prototype B.1.1 SARS-CoV-2 strain. The vaccine successfully passed preclinical studies [16] and Phase I/II clinical trials, showing good safety and immunogenicity profiles [17]. The CoviVac was certified in Russia for emergency use in mid-2021.

In the present study, the suitability and conditions of Size-Exclusion Chromatography (SEC) were investigated for the production of the human COVID-19 vaccine.

## 2. Materials and Methods

### 2.1. Cells and Viruses

The SARS-CoV-2 strain isolation, passage, and description was presented previously [16,18]. In brief, nasopharyngeal swab samples were collected from COVID-19 patients with fever (38–39 °C) and catarrhal symptoms. PCR-positive samples were used to isolate the viruses in Vero cells [18]. One of the isolated viruses obtained at the 3rd passage was used as a vaccine strain. The primary, main, and working seeds (namely passages 7, 8, and 9) were prepared and characterized in accordance with the national guidelines for the production of vaccines [16].

### 2.2. Preparation of Inactivated Virus Concentrate

Vero cells (obtained from WHO Biologicals (10–87)) were expanded on Cytodex 1 microcarriers (Cytiva, Marlborough, MA, USA) at 37 °C in EMEM (FSASI “Chumakov FSC R&D IBP RAS”, Moscow, Russia), and supplemented with fetal bovine serum (FBS, 10%). After reaching the confluent cell monolayers, cells were washed with Hanks buffer solution and the medium was exchanged to medium 199 (FSASI “Chumakov FSC R&D IBP RAS”, Moscow, Russia). The cultivation of the virus was carried out at 37 °C until complete degeneration of the Vero cell monolayer. Virus-containing suspension was collected and inactivated with β-propiolactone (1:1000 *v*/*v*). Inactivated supernatants were clarified using 0.65 + 0.45 μm filter (PES) and concentrated about 200-fold using cross-flow ultrafiltration (PES, 300 kDa cut-off). Therefore, inactivated concentrates contained mainly viral particles, host cell proteins, and nucleic acid.

### 2.3. Chromatography

An AKTA Pure (GE Healthcare Bio-Sciences, Uppsala, Sweden) was used for column chromatography. Media were packed into 26/20 (WorkBeads 40/100 (BioWorks, Uppsala, Sweden)) and 26/40 (Seplife 6 FF (SUNRESIN, Shaanxi, China), WorkBeads 40/1000 (BioWorks, Uppsala, Sweden), WorkBeads 40/10,000 (BioWorks, Uppsala, Sweden)) high-scale columns. A 1% (*v*/*v*) acetone solution was used for the test. Specifications of the columns are summarized in Table 1. Columns were decontaminated with 1v of 1 M sodium hydroxide after experiments. Columns were stored in 4% (*v*/*v*) formaldehyde at room temperature. The columns were equilibrated with phosphate buffer (40 mM, pH 7.2) containing 0.1 M NaCl to a stable conductivity value of 15 mS/cm. The load material (i.e., virus concentrates) was injected into 10 mL superloop (GE Healthcare Bio-Sciences, Uppsala, Sweden). Eluates were fractionated using the outlet valve of the chromatography system. SEC runs were conducted at a constant flow rate of 60 cm/h and a temperature of 20 °C using phosphate buffer (40 mM, pH 7.2) containing up to 0.1 M NaCl as eluent.

Purity was calculated using the following formula:$$P=\left(100-\frac{C_{TPF1}*V_{F1}}{C_{C}*V_{C}}*100\right)$$

*P*—purity, %;

*C*_*TPF*1_—concentration of total protein in the fraction, μg/mL;

*V*_*F*1_—fraction volume, mL;

*C*_*C*_—concentration of total protein in cell culture concentrate, μg/mL; and

*V*_*C*_—volume of the cell culture concentrate, mL.

### 2.4. Analytical Methods

#### 2.4.1. Total Protein Content Analysis

Total protein content in eluates was determined using the Lowry method [19] without precipitation [19,20].

#### 2.4.2. Viral Antigen Content Analysis with ELISA

Viral antigen content was quantified using an in-house indirect ELISA method. SEC eluate samples were diluted with PBS and incubated at 2–8 °C overnight on 96-well microtiter high-binding plates (Corning, Corning, NY, USA). After incubation, wells were washed 3 times with PBS-T (0.05%), then blocked with 2% BSA (FSBI «SCEEMP» of the Ministry of Health of the Russian Federation, Moscow, Russia) for 1h at 37 °C, and washed again 3 times. Diluted anti-SARS-CoV-2 rabbit serum IgG was then added to each well. These antibodies were purified from hyperimmune rabbit serum after immunization with inactivated SARS-CoV-2 via capture with protein-G resin (rProtien G FF, GE, Uppsala, Sweden). Furthermore, plates were incubated for 1h at 37 °C, washed 3 times, and incubated with HRP-conjugated mouse anti-rabbit antibodies (gamma-chain specific IgG, Sigma, St. Louis, MO, USA) for 1 h at 37 °C. Straight after the washing procedure, plates were stained with TMB substrate solution (Sigma), and incubated for 15 min at room temperature. The reaction was stopped with 0.5 M H_2_SO_4_. Optical densities were measured at 450 nm on a Multiskan SkyHigh Microplate Spectrophotometer (Thermo Fisher Scientific, Waltham, MA, USA). The data were analyzed using MS Excel.

#### 2.4.3. Host Cell DNA (hcDNA) Content Analysis

DNA was isolated from the studied samples (SEC eluates) using a set of reagents for the isolation of genomic DNA from biological material on the columns “K-SORB” (Syntol, Moscow, Russia). The concentration of Vero cell residual DNA was determined using quantitative PCR (qPCR) based on the TaqMan probe targeting actin gene (details and primers/probe sequences are available upon request).

#### 2.4.4. Bovine Serum Albumin (BSA) Content Analysis

The analysis for the BSA content was carried out using sodium dodecyl sulfate-polyacrylamide gel electrophoresis (SDS-PAGE) [21]. Samples and calibration solutions with known concentrations of BSA were separated in 12% gel and visualized by staining them with Coomassie brilliant blue R-250. BSA content was estimated in stained bands using GenSys software (Syngene, Frederick, MD, USA).

## 3. Results

Size-Exclusion Chromatography (SEC) was performed in separation mode (separation of the desired sample from the retained molecules by the resin pores). Columns packed with the following resins were tested: WorkBeads 40/100, 40/1000, and 40/10,000, and Seplife 6 FF; the characteristics of the resins are summarized in Table 1. Three rounds of purification were carried out for each resin, and three fractions (F1, F2, and F3) were collected on each round. Fractions F1 and F3 correspond to the peaks on the chromatograms, and F2 is the plateau between the peaks. For all chromatographic runs, the same virus concentrate was used. The concentrates were loaded onto columns at a 10% column volume (CV), as recommended by the manufacturers. The absorption of UV at 280 (red line) and 260 (blue line) nm was recorded continuously during purification. The chromatograms are shown in Figure 1. The obtained eluates were analyzed for the content of total protein, bovine serum albumin (BSA), residual cellular DNA, and specific viral antigen (Table 2).

Two clearly separated peaks were observed in the OD 280/260 elution profile chromatograms. The purest target product (viral antigen) was eluted through the dead volume of the column and collected in the first fraction (F1 on Figure 1). The second peak (F3 on Figure 1) contains impurities such as cellular and serum proteins, hcDNA, and BSA, the separation of which, by weight and size, did not occur. Coronavirus particles were found in both peaks. The viral antigen was found in F2 and F3 fractions along with high impurity contents (Table 2), which allowed us to conclude that these fractions were not suitable for further use. Therefore, the F1 fraction was chosen as the target one. However, the F1 fraction separated by WorkBeads 40/100 and WorkBeads 40/1000 contained detectable amounts of hcDNA and BSA. The characteristics of the F1 fraction, collected from columns with WorkBeads 40/10,000 and Seplife 6FF resins, were comparable.

The highest content of a specific viral antigen was observed when the concentrate was purified using WorkBeads 40/100 resin; however, a comparison of the elution profiles showed that target peaks of chromatograms from Seplife 6 FF resin columns were narrower compared to WorkBeads 40/100, 40/1000, 40/10,000, and the separation between the first and second peaks was much clearer (Figure 1A). Moreover, the purity of the F1 fraction was higher for Seplife 6FF resin (Table 3).

Based on the results of the analytical methods and the analysis of the elution profiles, we can conclude that on Seplife 6 FF resin, the purification of the concentrate from impurity proteins and residual cellular DNA is better than with other resins. Therefore, Seplife 6 FF resin can be recommended for the purification of inactivated coronavirus antigen.

The resins used herein for SEC were inert, unable to bind particles in the eluted sample, and differed only in the range of fractionated particles. The target size of the coronavirus particles is about 100–120 nm in diameter. The Seplife 6 FF resin significantly differed from the WorkBeads 40/100 and 40/1000 resins in pore size; the separation of viral particles on it showed better characteristics, and the resulting peak contained viral particles measuring 100–120 nm [16,22].

## 4. Discussion

SEC was used to separate small molecule impurities from inactivated coronavirus particles. Elutions from all four chromatography resins were similar and led to a characteristic double-peak pattern in the UV trace. The virus was eluted and collected in the first peak (void fraction), whereas the second peak resulted from the elution of smaller molecules (BSA, amino acids, nucleotides, etc.). Similar elution profiles have been reported for turkey corona virus (TCoV) propagated in the turkey embryo [14], despite the use of different chromatography media and feed material. In both studies, the virus was concentrated prior to injection.

Notably, some tailing of the virus (detected by ELISA) was observed upon elution from all resins. This finding suggests that part of the virus population (or deteriorated virions) was able to enter a fraction of the pores or bind with other components of concentrate media. In addition, mass transfer effects (due to the large size of virions) may have enhanced tailing of the elution peak.

HcDNA eluted in a plateau earlier than the host cell protein on average (Table 2), due to the size of hcDNA, or to binding with other components of the concentrate.

On average, low levels of extraction (from 7% to 40% for Seplife 6 FF, WorkBeads 40/100, WorkBeads 40/10,000, and WorkBeads 40/1000, respectively) of the target product can be explained by denaturation, aggregation, or the nonspecific adsorption of virions into the stationary phase.

## 5. Conclusions

The use of Seplife 6 FF resin for the SEC purification of a coronavirus antigen obtained from Vero cells makes it possible to obtain a target product purified from ballast components. SEC was found to be efficient for the separation of the virions from host cell proteins. Purity, however, must be balanced with efficiency in order for the process to be effective and economic. The purification from residual cell DNA is insufficient for the requirements of the Ph. Eur. (no more than 20 ng/mL in the final product) [23]; this requires further optimization of the existing purification method or the use of ion exchange chromatography as an additional purification step.

The proposed coronavirus purification method is fully scalable and does not require operations specific to a particular virus strain. Only standard operations (common for biotechnological processes) were used in this study, which makes the process suitable for large-scale industrial production.

## Figures and Tables

**Figure 1 vaccines-10-00949-f001:**
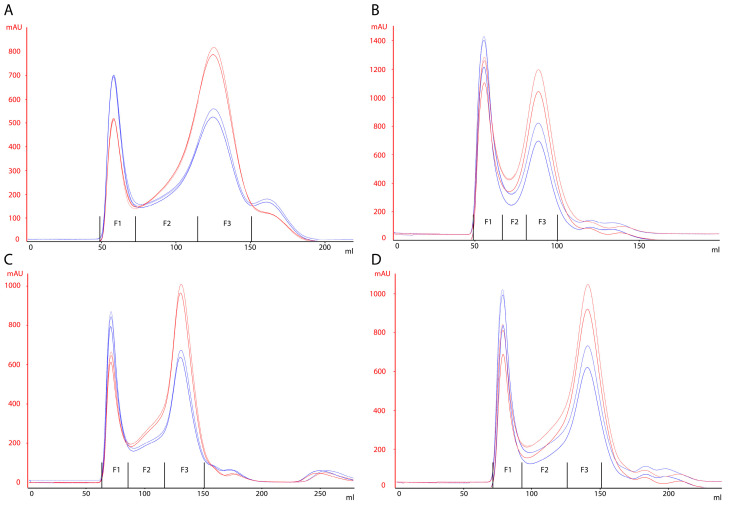
Size-Exclusion Chromatograms. Three rounds of purification were carried out for each resin, and three fractions (F1, F2, and F3) were collected on each round: (**A**) Seplife 6 FF, (**B**) WorkBeads 40/100, (**C**) WorkBeads 40/10,000, and (**D**) WorkBeads 40/1000.

**Table 1 vaccines-10-00949-t001:** Specifications of chromatography columns.

Resin	Exclusion Limit ^1^	H ^2^ (cm^−1^)	Vcol ^3^ (mL)	HETP ^4^ (mm)	Asym ^5^
Seplife 6 FF	10,000 kD ~4 × 10^6^ globulin	29.4	152	6345	1.35
WorkBeads 40/100	150 kD	18.4	98	11,955	0.98
WorkBeads 40/1000	1200 kD	34.0	180	9527	1.58
WorkBeads 40/10,000	10,000 kD	29.2	152	7996	2.09

^1^ Hydrodynamic diameter or molecular weight of globular protein excluded from the medium; ^2^ bed height; ^3^ column volume; ^4^ volume; ^5^ asymmetry of the elution peak for acetone.

**Table 2 vaccines-10-00949-t002:** Characteristics of the SEC fractions (data are presented as Mean ± SD; fractions 2 and 3 were only analyzed repeatedly for Seplife 6 FF resin).

Resin	Fraction	Total Protein, μg/mL	hcDNA, ng/mL	BSA, μg/mL	Viral Antigen, μg/mL
Virus concentrate	-	293,100	>100	>100	3900
Seplife 6 FF ^1^	F1	479 ± 115	23.3 ± 5.6	0	125 ± 39
	F2	1575 ± 526	45.0 ± 5.0	>10	206 ± 39
	F3	3780 ± 320	4.60 ± 2.65	>10	346 ± 127
WorkBeads 40/100	F1	2539 ± 108	55 ± 24	15.6 ± 15.4	721 ± 128
	F2	2851	6.51	>10	675
	F3	6679	1.42	>10	1391
WorkBeads 40/1000	F1	1307 ± 94	20.0 ± 3.6	4.6 ± 4.7	226 ± 37
	F2	1574	6.40	>10	304
	F3	5012	1.57	>10	1442
WorkBeads 40/10,000	F1	901 ± 45	0	>10	216 ± 52
	F2	1464	15.6	>10	447
	F3	5448	22.5	>10	1289

^1^*p* < 0.05 compared to WorkBeads 40/100, 40/1000 (unpaired, two-tailed Student’s *t*-test).

**Table 3 vaccines-10-00949-t003:** Purity of fraction F1.

Resin	Purity, %
Seplife 6 FF	99.63
WorkBeads 40/100	99.29
WorkBeads 40/1000	98.12
WorkBeads 40/10,000	99.03

## Data Availability

The data presented in this study are available within the article.

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
