# Peer review of "Methodology of Purification of Inactivated Cell-Culture-Grown SARS-CoV-2 Using Size-Exclusion Chromatography"

_vaccines, 2022, doi:10.3390/vaccines10060949_

Round 1
Reviewer 1 Report
The presented manuscript perfectly fits into the global trend of research on vaccines against SARS-CoV-2 virus. In the presented work, the authors check the suitability of of four size exclusion chromatography resins for purification of viral antigen for the human COVID-19 vaccine. The presented results and interesting and useful in terms of COVID-19 research. The manuscript can be accepted after minor revision.
- "2.1 Cells and Viruses": I think the Authors should resume isolation, passages and description very shortly. Otherwise reader has to check position [13] but in my opinion, it should be done only by someone really interested in copying/verifying the results. Typical read should have a fast view of the procedure in the paper.
- Table 1: when describing the Resin, please remove the company and country of origin. These should be placed directly in the text (2.3 Chromatography)
- Please improve the quality of the Figure 2. The colors should be uniformed and resolution and general quality improved.
- The Authors have to elaborate on the results in the conclusions: two sentences are not enough. Please compare your results with other techniques, what else can be done (any future experiments?)
Author Response
The presented manuscript perfectly fits into the global trend of research on vaccines against SARS-CoV-2 virus. In the presented work, the authors check the suitability of of four size exclusion chromatography resins for purification of viral antigen for the human COVID-19 vaccine. The presented results and interesting and useful in terms of COVID-19 research. The manuscript can be accepted after minor revision.
1. "2.1 Cells and Viruses": I think the Authors should resume isolation, passages and description very shortly. Otherwise reader has to check position [13] but in my opinion, it should be done only by someone really interested in copying/verifying the results. Typical read should have a fast view of the procedure in the paper.
Response: manuscript text was amended with a short description of the strain isolation, passage history and characterization.
2. Table 1: when describing the Resin, please remove the company and country of origin. These should be placed directly in the text (2.3 Chromatography)
Response: Corrected
3. Please improve the quality of the Figure 2. The colors should be uniformed and resolution and general quality improved.
Response: Corrected
4. The Authors have to elaborate on the results in the conclusions: two sentences are not enough. Please compare your results with other techniques, what else can be done (any future experiments?)
Response: Amended
Reviewer 2 Report
Please describe in more detail the conditions of size-exclusion chromatography.
In the introduction i suggest the authors to take some informations from this articles
DOI 10.3390/microorganisms8111704,
DOI
10.3390/microorganisms9040793,
DOI
10.2147/RMHP.S284557,
DOI
10.3390/antiox10060881
Could the authors please mention what are the differences from the other published article mentioned in the bibliography?
There is no discussion chapter, in the results and discussion chapter the authors presented just their results and not aalso discussions.
Author Response
Please describe in more detail the conditions of size-exclusion chromatography.
Response: The data on SEC conditions is presented in Section 2.3. Text was amended.
In the introduction i suggest the authors to take some informations from this articles
DOI 10.3390/microorganisms8111704,
DOI 10.3390/microorganisms9040793,
DOI 10.2147/RMHP.S284557,
DOI 10.3390/antiox10060881
Could the authors please mention what are the differences from the other published article mentioned in the bibliography?
Response: These manuscripts mainly describe the data and experience collected by clinical specialists during the peak of the COVID-19 pandemic. As our manuscript focuses on the technological process of inactivated coronavirus antigen purification, we decided not to overload the paper with clinical data on COVID-19 and included only main facts to put the research into context. The technology presented is not specific for SARS-CoV-2, and can be used for purification of other CoVs’ antigen.
There is no discussion chapter, in the results and discussion chapter the authors presented just their results and not aalso discussions.
Response: The Discussion was added and elaborated.
Reviewer 3 Report
we read with interest the article by Kovpak et al discussing the methodology of using size exclusion column chromatography for isolating and purifying the viral samples ( SARS-CoV-2). the article is written in a good way however; it has some major deficits that need to be answered.
First the article falls in the methodology category and thus the whole method section should be described in an analytical level in terms of amount of proteins loaded (mg) and describe fraction collection and recovery rate as well as purity to be assessed by Western blotting to show that the separation of size exclusion in the different fractions is achieving the size separation. These comments are related to the methodology section.
In terms of concepts, the authors do not describe what are they achieving specifically, they mention viral antigen, are they purifying the S protein and if so what is the molecular weight of the pure protein and what is the activity level of this protein after separation, all these data and experiments should be performed. the way it is described is vague and it is described as viral antigen?
In addition, the article describes a methodology of the size exclusion, the authors should state if there is a gap of knowledge in achieving and performing this method successfully and if there are challenges, if not, this report carries no scientific value.
The ELISA is described in a very simplistic method where the amount and time of incubation and antibodies concentrations and sources are not described also the results if these ELISA reading should be presented and the raw values should be presented as supplementary data.
The introduction should discuss previous work on protein separation using size exclusion and why the authors are presenting an article about it.
Author Response
we read with interest the article by Kovpak et al discussing the methodology of using size exclusion column chromatography for isolating and purifying the viral samples ( SARS-CoV-2). the article is written in a good way however; it has some major deficits that need to be answered.
First the article falls in the methodology category and thus the whole method section should be described in an analytical level in terms of amount of proteins loaded (mg) and describe fraction collection and recovery rate as well as purity to be assessed by Western blotting to show that the separation of size exclusion in the different fractions is achieving the size separation. These comments are related to the methodology section.
Response: The present manuscript describes only the process of the viral antigen purification. The characteristics of the purified antigen itself (SDS-PAGE and WB) are presented in [13].
In terms of concepts, the authors do not describe what are they achieving specifically, they mention viral antigen, are they purifying the S protein and if so what is the molecular weight of the pure protein and what is the activity level of this protein after separation, all these data and experiments should be performed. the way it is described is vague and it is described as viral antigen?
Response: In this manuscript, we investigated the applicability of gel filtration sorbents for the purification of inactivated whole-virion coronavirus particles obtained via reproduction in Vero cells. Therefore, the antigen is not a single viral protein S, but a whole viral particle capable of inducing an immune response.
Response: Introduction part was amended to put the present work into context.
The ELISA is described in a very simplistic method where the amount and time of incubation and antibodies concentrations and sources are not described also the results if these ELISA reading should be presented and the raw values should be presented as supplementary data.
Response: Materials and Methods section was supplemented with detailed description of the ELISA procedure. The whole procedure is fairly basic.
The introduction should discuss previous work on protein separation using size exclusion and why the authors are presenting an article about it.
Response: Introduction part was amended to put the present work into context.
Round 2
Reviewer 3 Report
Thank you for the clarification and I think the work would be much stronger if this work focused on the protein ID and the virology aspect of this work. In the present work it is a technical report of a methodology applying size exclusion separation.
I would request that the title be changed into: Methodology of Purification of Inactivated Cell Culture Grown SARS-CoV-2 by Size-Exclusion Chromatography
Author Response
The title has been changed as recommended